# Torrefaction of Sewage Sludge: Kinetics and Fuel Properties of Biochars

**Jakub Pulka [1,2]** , **Piotr Manczarski [3]** , **Jacek A. Koziel [4]** and **Andrzej Białowiec [2,\*]**

1  Faculty of Agronomy and Bioengineering, Poznan University of Life Sciences, 60-637 Poznań, Poland; jakub.pulka@up.poznan.pl

2  Faculty of Life Sciences and Technology, Wrocław University of Environmental and Life Sciences, 50-375 Wrocław, Poland

3  Department of Environmental Engineering, Faculty of Building Services, Hydro and Environmental Engineering, Warsaw University of Technology, 00-653 Warsaw, Poland; piotr.manczarski@pw.edu.pl

4  Department of Agricultural and Biosystems Engineering, Iowa State University, Ames, IA 50011, USA; koziel@iastate.edu

\*  Correspondence: andrzej.bialowiec@upwr.edu.pl; Tel.: +48-71-320-5973

**Abstract:** We propose a 'Waste to Carbon' thermal transformation of sewage sludge (SS) via torrefaction to a valuable product (fuel) with a high content of carbon. One important, technological aspect to develop this concept is the determination of activation energy needed for torrefaction. Thus, this research aimed to evaluate the kinetics of SS torrefaction and determine the effects of process temperature on fuel properties of torrefied products (biochars). Torrefaction was performed using high ash content SS at six (200~300 °C) temperatures and 60 min residence (process) time. Mass loss during torrefaction ranged from 10~20%. The resulting activation energy for SS torrefaction was ~12.007 kJ·mol$^{-1}$. Initial (unprocessed) SS higher heating value (HHV) was 13.5 MJ·kg$^{-1}$. However, the increase of torrefaction temperature decreased HHV from 13.4 to 3.8 MJ·kg$^{-1}$. Elemental analysis showed a significant decrease of the H/C ratio that occurred during torrefaction, while the O/C ratio fluctuated with much smaller differences. Although the activation energy was significantly lower compared with lignocellulosic materials, low-temperature SS torrefaction technology could be explored for further SS stabilization and utilization (e.g., dewatering and hygienization).

**Keywords:** waste to carbon; sewage sludge; kinetics; torrefaction; biochar; activation energy; fuel properties

## 1. Introduction

### 1.1. Sewage Sludge Management

The increasing demand for electricity and heat drives the development of new technologies for generating energy from so-called renewable resources. Waste streams (municipal, food industry, agricultural production) have growing importance that as a result of thermal or biological processes, can be converted into renewable energy. Processed waste can be used as a secondary raw material [1].

Sewage sludge (SS) is a waste material that is produced in large quantities worldwide. However, its utilization can be problematic due to high organic fraction content and moisture. Most developed countries ban SS landfilling. Agricultural usage is problematic due to possible toxicity and impact on groundwater, and incineration is expensive. Twenty-eight EU countries produced more than 10 million Mg of dry sewage sludge yearly [2]. Main disposal techniques include agriculture applications or landscaping, incineration and landfilling [3]. SS management differs significantly between countries; countries with the highest GDP prefer incineration as a main way of utilization, others are forced to

landfill. EU incinerates ~22% of total amount of SS, with the Netherlands and Switzerland incinerating all of the produced SS, Germany incinerates 56%, but Romania and Bosnia Herzegovina landfill almost all of the produced SS [4]. Poland generates > 0.5 million Mg·year$^{-1}$ (dry basis), of which ~56% is stored (in lagoons at wastewater treatment plants) for further utilization [5] which involves agricultural usage (38%), thermal transformation (13.5%), and landfilling (6%) [6]. With landfilling being the least desirable solution and incineration being rather expensive, there is still a need for new solutions to sustainable SS utilization. One of the novel approaches we propose is to produce biochar from SS using the torrefaction process [7].

### 1.2. Torrefaction

Torrefaction is a low-temperature biomass thermal decomposition process that produces a carbon-rich product—biochar [8]. Biomass partly decomposes during this process generating both condensable and non-condensable gases. The resulting product is a solid substance rich in carbon, referred to as 'biochar,' 'torrefied biomass' or 'biocarbon' [9]. In industry and literature, the torrefaction process is also referred to as 'roasting,' 'slow and mild pyrolysis,' 'wood-cooking,' and 'high-temperature drying' [10].

The torrefaction process improves several fuel properties of biomass. Energy density can increase by ~30% [11]. Torrefied biomass becomes hydrophobic [12]. The fixed C content of torrefied biomass is higher than the raw material [13]. Torrefaction reduces the O/C which makes the biomass better suited for gasification [14]. Torrefied biomass grindability is superior to that of raw biomass [15,16]. Torrefied biomass takes less time for ignition [17]. It is also possible to pelletize torrefied biochar [18].

Temperature and retention time are two main parameters that influence torrefaction process efficiency [19]. Torrefaction is usually conducted at temperatures between 200~300 °C [20], and the process temperature is maintained for 15 to 60 min [21]. Choosing the specific value of those two key parameters for different types of biomass is essential to develop cost-effective biomass treatment.

To date, the torrefaction process was mainly applied for lignocellulose biomass treatment [22]. Plants with the highest lignocellulose percentage compared to sugars and fats have the best energy potential [23]. Feedstock currently used in commercial scale or in research include wood chip and wood pellets, tree bark, crop residues (straw, nut shells and rice hulls), switchgrass, organic wastes including distillers' grain, bagasse from the sugarcane industry, olive mill waste, poultry litter, dairy cattle manure and paper sludge [24].

### 1.3. Sewage Sludge Torrefaction

Due to urbanization, production of SS has increased substantially, and the torrefaction process could provide a sustainable utilization of this large volume of non-lignocellulosic biomass/waste. However, torrefaction of low lignocellulosic content feedstock is less known [7,25,26]. Sewage sludge [7,25,26], digestate from biogas plants [27], agricultural animal waste [6], municipal solid waste [28] and food waste [29] contain fats, proteins and other organic matter with very low lignocellulose content.

There is a major gap in knowledge related to SS torrefaction. Basic research is still needed to explore the potential of implementing this technology. For example, the effects of SS initial properties, the SS origin, and a type of wastewater treatment needs to be explored. Systematic knowledge of SS torrefaction and transformation to biochar with desirable properties has not been developed yet.

To date, the only research on the kinetics of SS torrefaction was published by Poudel et al. [26] who used SS with very low (2.6% w.m.) ash content. This is in contrast to a greater number of studies focused on the kinetics of pyrolysis. Lessons learned from pyrolysis can be useful but are not necessarily applicable to SS torrefaction modeling and optimization.

The kinetics of torrefaction of high ash content SS is not explored yet. High ash content is generally not desirable in SS as it results in lower heating value. However, high ash content in SS is site/source and treatment technology specific. Sewage sludge (with ash content ranging from 12%

to 43% d.m.) was torrefied, resulting in biochar with HHVs of 11.43 to 19.86 MJ·kg$^{-1}$ [26,30–33]. Research on the effects of high ash content is required to build a database informing and enabling the SS torrefaction applicability to SS with a wider range of properties. Research on lignocellulosic biomass torrefaction [34–36] provides a good example of a useful database. Similarly, a gap in knowledge exists on SS torrefaction kinetics. This is in contrast to a relatively better-known kinetics of pyrolysis describing thermal processing in the higher temperature range [37,38].

*1.4. Objectives*

This study aimed to evaluate the torrefaction kinetics of SS with high initial ash content. The influence of process temperature on SS biochar fuel properties was also investigated.

## 2. Materials and Methods

*2.1. Sewage Sludge*

Sewage sludge was acquired from a municipal wastewater treatment plant (MWWTP) in Jastrzębie-Zdrój, Poland. This facility is a mechanical-biological wastewater treatment plant that receives 14,000 m$^3$ d$^{-1}$, which is a 148,145 population equivalent (PE). Treatment includes mechanical separation, biological treatment with aerobic, anaerobic and anoxic tanks. Wastewater is separated from SS in clarifiers, and SS is subsequently fermented and dewatered. Dewatered SS samples were periodically collected from 100 kg initial bulk sample from which 1 kg was stored in several 100 cm$^3$ vessels at −20 °C. SS properties are presented in Table 1.

**Table 1.** Prosperities of sewage sludge from MWWTP in Jastrzębie-Zdrój (results of 6 months of data gathered in 2016, mean ± standard deviation (SD)).

| Sewage Sludge Parameter | February | April | June | August | October | December | Mean ± SD |
|---|---|---|---|---|---|---|---|
| Ash content, % d.m. | 45.60 | 44.00 | 42.00 | 43.00 | 44.00 | 40.00 | 43.10 ± 1.93 |
| pH | 8.50 | 8.70 | 8.70 | 8.30 | 8.30 | 8.30 | 8.47 ± 0.20 |
| Dry mass, % | 18.40 | 19.00 | 19.00 | 19.00 | 19.00 | 16.00 | 18.40 ± 1.20 |
| Organic matter, % d.m. | 54.40 | 56.00 | 58.00 | 53.00 | 56.00 | 60.00 | 56.23 ± 2.50 |
| Ammonium nitrogen, % d.m. | 0.20 | 0.77 | 0.27 | 0.37 | 0.34 | 0.37 | 0.39 ± 0.20 |
| Total nitrogen, % d.m. | 3.73 | 4.10 | 4.20 | 4.20 | 3.90 | 4.30 | 4.07 ± 0.22 |
| Total phosphorus, % d.m. | 1.64 | 3.10 | 2.20 | 3.40 | 3.30 | 2.80 | 2.74 ± 0.69 |
| Lead (Pb), mg·kg d.m. | 20.00 | 23.00 | 18.00 | 18.00 | 17.00 | 17.00 | 18.83 ± 2.32 |
| Cadmium, (Cd) mg·kg$^{-1}$ d.m. | 1.20 | 1.40 | 1.10 | 1.20 | 1.10 | 0.99 | 1.17 ± 0.14 |
| Copper (Cu), mg·/kg$^{-1}$ d.m. | 104.00 | 116.00 | 117.00 | 124.00 | 127.00 | 121.00 | 118.17 ± 8.08 |
| Nickel (Ni), mg·kg$^{-1}$ d.m. | 14.70 | 16.00 | 14.00 | 15.00 | 17.00 | 15.00 | 15.28 ± 1.06 |
| Mercury (Hg), mg·kg$^{-1}$ d.m. | 0.93 | 0.86 | 2.20 | 1.20 | 0.59 | 0.78 | 1.09 ± 0.58 |
| Zinc (Zn), mg·kg$^{-1}$ d.m. | 559.00 | 653.00 | 632.00 | 583.00 | 624.00 | 567.00 | 603.00 ± 38.51 |

*2.2. Experimental Apparatus*

The experimental torrefaction unit used for the SS was described in detail by Stępień et al., [39]. To ensure that the inert atmosphere was maintained, the CO$_2$ gas was introduced from the bottom of the reactor at a rate of 0.6 mL·min$^{-1}$. The investigated SS sample was placed in the cuvette and introduced inside the reactor. The cuvette was integrated with the electronic balance with 0.01 g resolution to enable the measurement of the mass loss during the torrefaction process. The parameters of the torrefaction process were registered by a PC and exported to the file.

*2.3. Torrefaction*

All samples were first dried at 105 °C. Approximately 2.25 g of dried SS samples were placed in the reactor and heated at different constant temperatures of 200, 220, 240, 260, 280, and 300 °C for up to 1 h. Temperature range and intervals were typical for torrefaction temperatures [10] and based on the methodology described by Bialowiec et al. [28].

### 2.4. Sewage Sludge and Biochar Analyses

A laboratory drier (model KBC-65W, WAMED, Warsaw, Poland) was used to determine moisture content according to standard PN-EN 14346:2011. Lost on ignition was determined using muffle furnace (model 8.1/1100, SNOL, Utena, Lithuania) according to standard PN-EN 15169:2011. The same furnace was used to determine combustible content according to standard PN-EN ISO 18122:2016-01. C, H, N, O, S elemental content analysis was measured with the elemental CHNS analyzer (CE Instruments Ltd., Manchester, UK). Sulfur was determined by the atomic emission spectrometry method with excitation in inductively coupled plasma (ICP-AES) after microwave mineralization using the atomic emission spectrometer (iCAP 7400 ICP-OES, Thermo Fisher Scientific, Waltham, MA, USA).

The O content was determined by the calculation method according to Equation (1):

$$O = 100\% - C - H - N - S - Ash \tag{1}$$

where: $O$ = oxygen content, %, $C$ = carbon content, %, $H$ = hydrogen content, %, $N$ = nitrogen content, %, $S$ = sulfur content, %, $Ash$ = ash content, %.

Higher heating value was measured using calorimeter (model C 200, IKAPOL, Warsaw, Poland) according to standard ISO 1928:2009. The lower heating value (LHV) was determined according to PN-Z-15008-04:1993 by calculation method, based on Equation (2):

$$LHV = HHV - r \cdot (W + 8.94 \cdot H) \tag{2}$$

where: $LHV$ = lower heating value, J·g$^{-1}$, $HHV$ = higher heating value, J·g$^{-1}$, $r$ = heat of water evaporation, 24.42 J·g$^{-1}$ for each 1% of water in fuel, J·g$^{-1}$, $W$ = moisture content, %, 8.94 = conversion factor of hydrogen to water, no units, $H$ = hydrogen content, %.

### 2.5. Data Analyses

The torrefaction rate constant was evaluated using the first-order reaction rate model [34]:

$$M_s = M_s^0 \times exp^{-k \times t} \tag{3}$$

$$ln\, M_s^0 / M_s = k \times t \tag{4}$$

where: $M_s^0$ = initial SS mass, g, $M_s$ = unit mass, g, $k$ = torrefaction rate constant, s$^{-1}$, $t$ = time, s.

The Arrhenius equation [28] describes the dependence of torrefaction rate constant ($k$) and temperature ($T$):

$$k(T) = A exp - E_a / RT \tag{5}$$

and in logarithmic form:

$$\ln k(T) = ln A - E_a / RT \tag{6}$$

where: $R$ = gas constant, 8.314, J·(mol·K)$^{-1}$, $T$ = temperature, K, $A$ = frequency factor, $E_a$ = activation energy J·mol$^{-1}$, $k$ = torrefaction rate constant, s$^{-1}$.

The activation energy can be calculated using the torrefaction rate constant using and the Arrhenius equation. Ln ($k$) is a linear function of 1/T [28]:

$$y = ax + b \tag{7}$$

where: $y = \ln(k)$, $b = ln A$, $a = E_a / R$

Dry mass loss was calculated with Equation (8) [29]:

$$My = \left( M^0 - M^1 / M^0 \right) \times 100\% \tag{8}$$

where: $M^0$ = dry mass before the process, g, $M^1$ = dry mass after the process, g.

## 3. Results and Discussion

### 3.1. Sewage Sludge Mass Loss during Torrefaction

Table 2 summarizes the mass loss and the percent mass loss during torrefaction for all six temperatures tested. Mass loss results obtained from SS (with high initial ash content) torrefaction at 300 °C, are much greater than those obtained at lower temperatures (Table 2). SS mass decrease ranged from 10% at 200 °C to 21% at 300 °C, which is lower than the results reported in [30,34]. Low % mass loss in this research is also lower compared with [26] where dry SS was torrefied for 50 min, and mass loss ranged from 75% to 55% for samples torrefied at 250 and 300 °C, respectively. Thus, SS characteristics have to be taken into account because, although SS is rather homogenous material, it differs between facilities it originates from, due to the difference in wastewater composition and the wastewater treatment setup.

**Table 2.** Mass after sewage sludge torrefaction and percentage mass loss (Equation (8)) for six temperature profiles (200, 220, 240, 260, 280, 300 °C).

| Torrefaction Temperature, °C | Mass after Torrefaction, g | Mass Loss, % |
|:---:|:---:|:---:|
| 200 | $2.03 \pm 0.010$ | $10\% \pm 0.44$ |
| 220 | $2.05 \pm 0.018$ | $9\% \pm 0.81$ |
| 240 | $2.01 \pm 0.017$ | $11\% \pm 0.77$ |
| 260 | $1.98 \pm 0.018$ | $12\% \pm 0.79$ |
| 280 | $1.95 \pm 0.023$ | $13\% \pm 1.02$ |
| 300 | $1.81 \pm 0.040$ | $20\% \pm 1.79$ |

In summary, torrefaction has the potential to decompose and volatilize solids that are present in SS. Lower mass loss during torrefaction in this research compared to the results shown in [26,30] and in [40], where oxidative torrefaction was studied, could be considered as a positive outcome when evaluating fuel potential of biochar from SS. Results in this study are comparable to those reported for mass loss during torrefaction of lignocellulosic biomass (10%~30%, 1 h torrefaction) [41] depending on the type of wood used.

### 3.2. Sewage Sludge Torrefaction Kinetics

Sewage sludge torrefaction rate constant ranged between $2.82\cdot10^{-5}$ to $6.71\cdot10^{-5}$ s$^{-1}$ (Table 3). The increasing temperature gradually increased the torrefaction rate constant with the exception of sample torrefied at 220 °C (Table 3). The activation energy of SS was ~12 kJ·mol$^{-1}$ (Table 3). Lim et al., [42] and Poudel et al., [26] described higher activation energy at, valued at 78.7 and 70.1 kJ·mol$^{-1}$, respectively. Similarly, in the work of Xu et al. [36] activation energy of SS pyrolysis 23 kJ·mol$^{-1}$ was higher but more similar to obtained in the present study. Higher activation energy may be attributed to smaller sample mass, higher range of temperatures, as well as lower initial ash content.

**Table 3.** Kinetic parameters of sewage sludge torrefaction.

| Temperature, °C | Torrefaction Constant Rate ($k$) Mean $\pm$ SD, s$^{-1}$ | Determination Coefficient ($R^2$) | $1/T$ (K$^{-1}$) | ln($k$) | Activation Energy, J·mol$^{-1}$ |
|:---:|:---:|:---:|:---:|:---:|:---:|
| 200 | $4.02\cdot10^{-5} \pm 3.00\cdot10^{-6}$ | 0.921 | 0.002114 | $-10.122$ | |
| 220 | $2.82\cdot10^{-5} \pm 6.26\cdot10^{-6}$ | 0.918 | 0.002028 | $-10.475$ | |
| 240 | $4.50\cdot10^{-5} \pm 4.76\cdot10^{-6}$ | 0.863 | 0.001949 | $-10.008$ | 12,007.91 |
| 260 | $4.57\cdot10^{-5} \pm 2.99\cdot10^{-6}$ | 0.873 | 0.001876 | $-9.993$ | |
| 280 | $4.29\cdot10^{-5} \pm 6.46\cdot10^{-6}$ | 0.845 | 0.001808 | $-10.056$ | |
| 300 | $6.71\cdot10^{-5} \pm 8.20\cdot10^{-6}$ | 0.912 | 0.001745 | $-9.610$ | |

The activation energy for lignocellulosic materials torrefaction can also be used for comparison. The torrefaction activation energy of pine and spruce was 165.0 kJ·mol$^{-1}$ and 103.8 kJ·mol$^{-1}$, respectively [35,43]. Sarvaramini et al., [44] evaluated torrefaction activation energy of cellulose and lignin to be ~56.3 kJ·mol$^{-1}$ and ~190.8 kJ·mol$^{-1}$ respectively, much higher than those obtained for the SS samples in this research. This finding is also consistent with the lower SS pyrolysis activation energy of 250 kJ·mol$^{-1}$ [45] compared to 388 kJ·mol$^{-1}$ for some types of wood biomass [46]. Higher activation energy (80 kJ·mol$^{-1}$) was also obtained for food waste [29] and was likely attributed to much higher mass loss caused by greater organic matter content.

A conclusion can be drawn that torrefaction of SS does not require a significant portion of energy to initiate the process compared to torrefaction of lignocellulosic material. This is likely due to the impact of lignocellulose content on high activation energy value [47]. Dry mass of SS contains mostly bacteria from activated sludge used in wastewater treatment, and its organic matter content ranged from 54 to 60% (Table 1). In contrast to wood biomass composed mainly of 45% cellulose, 5% lignin and 20% hemicellulose [48], organic matter in sewage sludge contains three main groups of compounds - lipids (25%), proteins (45%) and carbohydrates (15%) [49]. It is likely that torrefaction activation energy for these aforementioned groups of compounds is mostly related to volatilization, denaturation, caramelization as opposed to degradation. Therefore, it could be hypothesized that those major groups of compounds present in sewage sludge are much more susceptible to thermal treatment compared with lignocellulosic materials.

### 3.3. Torrefied Sewage Sludge Fuel Properties

Elemental analysis (Table 4) illustrates the trend of decreasing C, H, N, O content with the increased process temperature, while S concentration gradually increased. C, H, N, O loss was not entirely incremental and had some exceptions at 220 and 240 °C. H/C molar ratio decreased gradually with the temperature increase except for 220 and 280 °C temperature profiles (Table 4). H/C and O/C molar ratios are important fuel parameters that could indicate material incineration susceptibility. H/C molar ration results obtained by Poudel et al., [26], show a similar decreasing trend (from 1.75 to 1.63) with increasing temperatures compared to the observed decrease in this research from 1.94 to 0.67. The difference could be explained by the shorter residence time used by Poudel et al., [26]. Similarly, in the work of Atienza-Martínez et al. [32] both described molar ratios decreased in especially in the process carried out at 270 °C and a short 6 min residence time. Pulka et al., [40] reported a similar trend in decreasing H/C ratio (from 1.3 to 0.08). The O/C ratio shown in Poudel et al., [26], decreased gradually, while results obtained in this research fluctuated, and no obvious trend was observed (Table 4). Initial, high ash content increased from 40% in dried SS to 73 % obtained via torrefaction at 300 °C. The same trend can be observed in Poudel et al., [26], but values differ significantly from 15% in dry SS to 28% obtained at 300 °C.

**Table 4.** Elemental analysis of raw sewage sludge and torrefied sewage sludge biochars at different temperatures.

| Raw SS/SS Biochar | C% | H% | N% | S% | O% | H/C | O/C |
|---|---|---|---|---|---|---|---|
| Raw SS | 29.7 | 4.81 | 4.02 | 0.12 | 21.08 | 1.94 | 0.53 |
| SS biochar 200 °C | 28.4 | 2.43 | 4.03 | 0.24 | 21.31 | 1.03 | 0.56 |
| SS biochar 220 °C | 31.7 | 3.47 | 4.21 | 0.32 | 16.54 | 1.31 | 0.39 |
| SS biochar 240 °C | 29.5 | 2.76 | 4.03 | 0.41 | 19.23 | 1.12 | 0.49 |
| SS biochar 260 °C | 17.1 | 1.43 | 3.24 | 1.41 | 13.13 | 1.00 | 0.57 |
| SS biochar 280 °C | 11.3 | 1.27 | 2.63 | 1.50 | 11.01 | 1.35 | 0.73 |
| SS biochar 300 °C | 12.7 | 0.71 | 2.68 | 1.46 | 9.02 | 0.67 | 0.53 |

Initial HHV of dry SS was 13.503 MJ·kg$^{-1}$, and then it decreased during torrefaction, with highest declines observed above 260 °C, with the lowest HHV obtained at 300 °C (3.881 MJ·kg$^{-1}$) (Table 5). Poudel et al., [26] described a similar trend of decreasing HHV with temperature except for the

decrease rate that can be observed at higher torrefaction temperatures (Table 5). The relatively rapid HHV decrease observed at temperatures above 260 °C in this research may be associated with the degradation of proteins and soluble saccharides as previously described in [50]. In the work of Atienza-Martínez et al. [32], a decrease in HHV can be seen in samples torrefied at 320 °C after just 10 min of the process. SS used in this research had a much lower initial HHV value and significantly higher ash content compared to Poudel et al., [26] and Pulka et al., [40]. The additional difference was that in present research dried SS was torrefied for 60 min. In the case of Pulka et al., [40], wet SS was torrefied for 60 min. In our previous work [40], the constant level of biochars' HHV was achieved for all temperatures between 200~300 °C. It could be caused by shorter torrefaction time due to that, part of residence time was used for water removal.

**Table 5.** Moisture, volatiles, ash content, HHV, and LHV of sewage sludge and torrefied sewage sludge biochars at different temperatures.

| Raw SS/SS Biochar | Moisture, % | Volatiles, % | Ash, % | HHV, MJ·kg$^{-1}$ | LHV, MJ·kg$^{-1}$ |
|---|---|---|---|---|---|
| Raw (dry) SS | 7.1 | 59.7 | 40.3 | 13.503 | 12.279 |
| SS biochar 200 °C | 2.2 | 56.4 | 43.6 | 12.950 | 12.366 |
| SS biochar 220 °C | 3.0 | 56.2 | 43.8 | 13.204 | 12.373 |
| SS biochar 240 °C | 2.0 | 55.9 | 44.1 | 13.403 | 12.752 |
| SS biochar 260 °C | 3.9 | 36.3 | 63.7 | 6.503 | 6.095 |
| SS biochar 280 °C | 2.0 | 27.7 | 72.3 | 4.088 | 3.762 |
| SS biochar 300 °C | 2.6 | 26.6 | 73.4 | 3.881 | 3.663 |

Biochar properties reported in this research indicate that SS with high ash content is not suited for incineration as a fuel. However, the resulting LHV confirmed that the torrefaction temperature should not exceed 240 °C if the purpose of the proposed SS torrefaction is the production of solid fuel (Table 5). It is important to point out that raw SS parameters can greatly vary depending on the wastewater properties, treatment plant process, and SS management methods. Further studies should consider using different types of sewage sludge to develop a database on the torrefaction process, including the influence of sewage sludge initial properties on the torrefaction mass and energy balances and biochar properties. This knowledge will be useful for engineers to select technological parameters and decision makers to select a sustainable purpose of torrefaction and the resulting biochar utilization. Further investigation, with scaling up the torrefaction technology for SS treatment, should be continued.

## 4. Conclusions

We determined the torrefaction kinetics of sewage sludge with high initial ash content. The effect of torrefaction temperature on SS biochar fuel properties was tested. Several main conclusions can be made:

- The activation energy (12 kJ·mol$^{-1}$) of sewage sludge with high ash content in the torrefaction process was ~6-fold lower than reported for sewage sludge with 3~4-fold lower ash content. Further investigation on the influence of ash content of sewage sludge kinetics should be continued;
- Torrefaction temperature above 260 °C resulted in rapid organic matter loss, decreased HHV and LHV to values >6.5 and 6.1 MJ·kg$^{-1}$, respectively, and increased ash content >64%. SS with high initial ash content is not suitable for the production of quality fuel when torrefaction temperatures are >240 °C.
- Low-temperature torrefaction (< 240 °C) might still be useful for further sewage sludge (with high ash content) stabilization (e.g., dewatering and hygienization), and consideration for energy recovery from resulting biochar.

**Author Contributions:** Conceptualization, A.B., P.M.; methodology, A.B., J.P., P.M.; formal analysis, A.B., J.P., J.A.K.; validation, J.P., A.B., J.A.K.; investigation, J.P.; resources, J.P., A.B., P.M.; data curation, A.B., J.A.K.; writing—original draft preparation, J.P., A.B.; writing—review and editing, A.B., P.M., J.A.K.; visualization, J.P., A.B., J.A.K.; supervision, A.B., P.M., J.A.K.

**Funding:** The research was supported by the Preludium 6 Program, the National Research Center, Poland, grant #UMO-2013/11/N/NZ9/04587 titled "The influence of sewage sludge torrefaction on biocarbon phytotoxicity, leachability of heavy metals from biocarbon and sorption parameters of biocarbon." Authors would like to thank the Fulbright Foundation for funding the project titled "Research on pollutants emission from Carbonized Refuse Derived Fuel into the environment," completed at the Iowa State University. In addition, this paper preparation was partially supported by the Iowa Agriculture and Home Economics Experiment Station, Ames, Iowa. Project no. IOW05556 (Future Challenges in Animal Production Systems: Seeking Solutions through Focused Facilitation) sponsored by Hatch Act and State of Iowa funds. The publication is financed under the program of the Minister of Science and Higher Education; Strategy of Excellence—University of Research; in 2018–2019 project number 0019/SDU/2018/18 in the amount of PLN 700 000".

**Conflicts of Interest:** The authors declare no conflict of interest. The funders had no role in the design of the study; in the collection, analyses, or interpretation of data; in the writing of the manuscript, or in the decision to publish the results.

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
