# Peer review of "Torrefaction of Sewage Sludge: Kinetics and Fuel Properties of Biochars"

_energies, doi:10.3390/en12030565_

Round 1
Reviewer 1 Report
Title:
Torrefaction of sewage sludge: kinetics and fuel properties of biochars
Recommendation:
Major revision is required on this manuscript.
Comments:
This study evaluated the kinetics of sewage sludge torrefaction and determine the effects of process temperature on fuel properties of biochar. The topic of this research is interesting and timely. Some questions and suggestions are offered below with the intent to assist the author in improving the manuscript.
1. First of all, what year is the month in Table 1? Does these data have references and statistical values?
2. Line 101, author should add the country name, for example “Poland” after the city name.
3. The torrefaction temperatures in the study are 200, 220, 240, 260, 280, and 300 °C. What is the reason for choosing these six temperatures in this interval? Please give more information and explanation on this part.
4. References should be recheck, there are some format errors. For example, some publish years are not bold.
5. English expression needs to be improved and compact description of academic in whole manuscript is warranted.
Author Response
We included our reply in the attached file.

Reviewer 2 Report
The reviewer see that the authors made a significant efforts to prepare this manuscript under the title “Torrefaction of sewage sludge: kinetics and fuel properties of biochars”. The reviewer believes, the most important part of any research is the “results and discussion” section. The authors were properly prepared this section.
General comments:
It is clear that the literature review of this study made almost five years ago (see, the list of bibliography). For example, they were cited only few studies (six studies) from 2015-2019. Therefore, the whole paper need to be adjusted for the current time (~ February 2019), namely the background of this study. There are many recent studies focused on the same area. The authors were failed to prove the novelty of this study. The authors cannot prove the novelty of their work without the recent literature.
The authors only focus on the background of this study in the introduction section. There are not any explanations about the work itself. In addition, the objectives of this study need to be connected with the scientific gap that needs to be done. They presented the objective of this study in the individual section without knowing the reason.
The conclusions should reflect the aim of the paper. Please focus on the aim and the significant findings. Try to present them in bullets and avoid the specific results. I suggest to present them in bullets.
Specific comments
The reviewer suggest to avoid abbreviation in the title of any sections (e.g. SS torrefaction kinetics, SS mass loss during torrefaction).
Author Response

(The authors gave the same response as above.)

Round 2
Reviewer 1 Report
Thank you very much for reply.
I am pleased to tell you that your work can now be accept for publication in Energies.
Best wishes
Reviewer 2 Report
The authors were considered all the reviewer’s comments.